# RNA-Binding Protein MAC5A Is Required for Gibberellin-Regulated Stamen Development

**DOI:** 10.3390/ijms23042009

**Published:** 2022-02-11

**Authors:** Hua Liu, Hongna Shang, Huan Yang, Wenjie Liu, Daisuke Tsugama, Ken-Ichi Nonomura, Aimin Zhou, Wenwu Wu, Tetsuo Takano, Shenkui Liu

**Affiliations:** 1State Key Laboratory of Subtropical Silviculture, Zhejiang A&F University, Hangzhou 311300, China; 17853729241@163.com (H.S.); yh18434762841@163.com (H.Y.); wwwu@sibs.ac.cn (W.W.); 2College of Agriculture, Northeast Agricultural University, Harbin 150030, China; a1975595206@sina.com; 3Asian Natural Environmental Science Center (ANESC), The University of Tokyo, Tokyo 188-0002, Japan; dtsugama@anesc.u-tokyo.ac.jp (D.T.); takano@anesc.u-tokyo.ac.jp (T.T.); 4Plant Cytogenetics Laboratory, National Institute of Genetics (NIG), Shizuoka 411-0801, Japan; knonomur@nig.ac.jp; 5Department of Life Science, Graduate University for Advanced Studies/Sokendai, Shizuoka 411-0801, Japan; 6College of Horticulture and Landscape Architecture, Northeast Agricultural University, Harbin 150030, China; aiminzhou@neau.edu.cn

**Keywords:** *Arabidopsis thaliana*, floral organ development, gibberellin, stamen, male sterility, MAC5A

## Abstract

The development of floral organs is coordinated by an elaborate network of homeotic genes, and gibberellin (GA) signaling is involved in floral organ development; however, the underlying molecular mechanisms remain elusive. In the present study, we found that MOS4-ASSOCIATED COMPLEX 5A (MAC5A), which is a protein containing an RNA-binding motif, was involved in the development of sepals, petals, and stamens; either the loss or gain of *MAC5A* function resulted in stamen malformation and a reduced seed set. The exogenous application of GA considerably exacerbated the defects in *mac5a* null mutants, including fewer stamens and male sterility. *MAC5A* was predominantly expressed in pollen grains and stamens, and overexpression of *MAC5A* affected the expression of homeotic genes such as *APETALA1* (*AP1*), *AP2*, and *AGAMOUS* (*AG*). MAC5A may interact with RABBIT EARS (RBE), a repressor of *AG* expression in *Arabidopsis* flowers. The petal defect in *rbe* null mutants was at least partly rescued in *mac5a rbe* double mutants. These findings suggest that *MAC5A* is a novel factor that is required for the normal development of stamens and depends on the GA signaling pathway.

## 1. Introduction

Gibberellins (GAs) are a family of tetracyclic diterpenoid phytohormones that stimulate plant growth and development, including flower development and inflorescence meristem size [1,2,3,4,5,6]. Flowers are frequently composed of four different floral organs: sepals, petals, stamens, and carpels; and the A, B, and C class genes of the ABC model are critical for floral organ identity [7,8]. The identity of sepals in the outer whorl is determined by class A genes *APETALA1* (*AP1*) and *AP2*, while that of petals in the second whorl is determined by the combined activity of class A and B genes *AP3* and *PISTILLATA* (*PI*). Stamens in the third whorl are determined by the combined effects of class B genes and the class C gene *AGAMOUS* (*AG*), which is also solely responsible for the determination of carpels in the innermost whorl. The anthers, which develop in the third whorl, are suggested to be a major site of bioactive GA synthesis during flower development in several plant species [9,10,11]. GA-deficient *ga1**-3* mutants develop flowers with retarded growth of all floral organs despite their normal identities, especially impaired another development, which results in male sterility owing to a lack of mature pollen [2,12,13,14]. Short anthers and complete male sterility in *ga1-3* mutants can be restored by exogenous GA application, which upregulates the expression of *AP3*, *PI*, and *AG* [2,15,16]. The overexpression of *AG* partially rescues floral defects in young flowers of *ga1-3* mutants [3].

GA promotes floral organ development in *Arabidopsis*
*thaliana* by blocking the functions of DELLA proteins [2]. DELLA proteins have an N-terminal with two conserved domains: the DELLA and TVHYNP domains. After the binding of bioactive GA, the C3-hydroxyl group of the GA molecule becomes hydrogen-bound to the GA receptor GIBBERELLIN INSENSITIVE DWARF1 (GID1), inducing a conformational change of GID1 in the N-terminal extension to cover the GA pocket [17,18]. Once the pocket is closed, the upper surface of the lid binds to the DELLA and TVHYNP domains of the DELLA protein to form a GA-GID1-DELLA complex [12,19,20]. Subsequently, the GA-GID1-DELLA complex is recognized by the SCFSLY1/GID2 ubiquitin E3 ligase complex for polyubiquitination, and the DELLA protein is then degraded by the 26S proteasome [21,22]. A reduction in GA concentration promotes the accumulation of DELLA proteins that repress GA responses, whereas an increase in GA results in the degradation of DELLA proteins and the activation of GA responses. Five *DELLA* genes (*GAI*, *RGA*, *RGL1*, *RGL2*, and *RGL3*) are present in the *Arabidopsis* genome. The loss of function of *RGA* and *RGL2* is almost sufficient to compensate for floral organ deficiency in *ga1-3* mutants, which indicates that both *RGA* and *RGL2* are major repressors of GA response in this process [21].

MOS4-ASSOCIATED COMPLEX 5A (MAC5A) is involved in the stress response, cell division, and development of *Arabidopsis*. In yeast and humans, MAC5A homologs are components of the NINETEEN COMPLEX (NTC) or Prp19 Complex (Prp19C) [23]. MAC/NTC/Prp19 is a large flexible RNA-protein complex associated with the spliceosome that permits alternative splicing and generates multiple transcript variants from a single gene, and it is required for development and immunity in all three organisms. The core components of MAC include CELL DIVISION CYCLE5 (CDC5/MAC1), WD-40 protein PLEIOTROPIC REGULATORYLOCUS1 (PRL1/MAC2), MAC3A and MAC3B, and MOS4 [23]. *MOS4* encodes a nuclear protein that binds directly to CDC5, an atypical R2R3 Myb transcription factor required for optimized Pol II activity at the promoters of miRNA genes [24]. During transcription, PRL1 binds to pri-miRNAs to prevent their degradation [25]. MAC3A and MAC3B interact physically and genetically with the Ski-interacting protein and mediate the alternative splicing of 50% of the expressed genes in *Arabidopsis* [26]. *mac5a* null mutants exhibit early flowering and have serrated leaves, short roots, and reduced seed sets [27]. The loss of function of *MAC5A* and its close homolog *MAC5B* is lethal, and *mac5a* homozygous plants with a *mac5b* heterozygous background exhibit defects in reproductive organogenesis [27]. In this study, we found that the overexpression of *MAC5A* and loss of *MAC5A* function affected stamen development. Furthermore, *MAC5A* predominantly expressed in anthers and regulated anther development via a GA-dependent pathway.

## 2. Results

### 2.1. Loss of MAC5A Function Affects Floral Organ Development in Arabidopsis

Salk_132281 (*mac5a*) carries a T-DNA insertion in the second exon of *MAC5A* and is a genuine *mac5a* null mutant, as previously reported [27]. In the present study, we showed that the petals of *mac5a* single mutants were narrower than those of the Columbia ecotype (Col-0) of *Arabidopsis* (Figure 1A,B,D). A normal *Arabidopsis* flower has two short and four long stamens, all of which develop in the third whorl (Figure 1B). *mac5a* had a reduced number of stamens (Figure 1B–D); 42% of *mac5a* flowers did not have any short stamens and produced four long stamens only, whereas 36% had five long stamens (Figure 1B,C). The outer epidermal cells of sepals exhibit a characteristic pattern with diverse sizes, ranging from giant cells of an average length of 360 μm to smallest cells only approximately 10 μm in length [28]. In this study, the Col-0 sepals developed a group of tightly arranged small cells to form a smooth edge (Figure 1E). In *mac5a*, the number of small cells was reduced, and giant cells frequently developed on the edge of the sepals, forming a jagged tip (Figure 1E). Compared with Col-0, *mac5a* exhibited irregularly shaped and collapsed pollen grains (Figure 1F), reduced seed set (Figure 1G–I), and short siliques (Figure 1J).

### 2.2. GA Application Enhances the Deficient of Stamen Development in mac5a Mutant

GA treatment exacerbated the abnormalities in the floral organs of *mac5a* null mutants, whereas PAC treatment partially alleviated them (Figure 2). Jagging of the sepal tip in *mac5a* mutants seemed to be aggravated by GA but was not recovered by PAC (Figure 2A). Following GA application, the number of stamens decreased (Figure 2B), and anther development was retarded in *mac5a*, resulting in a lack of mature pollen (Figure 2C). Pollen viability was assessed by Alexander cytoplasmic staining. In the presence of 100 μM GA_3_, around 70% of anthers in Col-0 developed viable pollen (Figure 2D,E). In contrast, pollen development was consistently aborted or invisible in *mac5a* anthers in response to GA treatment (Figure 2D,E), leading to sterility (Figure 2F). After PAC treatment, the number of stamens in *mac5a* increased and was comparable to that in Col-0 (Figure 2B).

### 2.3. Overexpression of MAC5A Affects Stamen Development in Arabidopsis

To gain further insights into the physiological role of *MAC5A*, a full-length *MAC5A* cDNA was fused in-frame with the *GFP* gene at the 3′-terminus and driven by the constitutive CaMV35S to overexpress the fusion gene *MAC5A-GFP* in *Arabidopsis* (ox lines). Eleven independent *MAC5A**-*overexpressing lines were identified. The T_2_ populations of ox2, ox4, ox8, and ox11 segregated abnormal phenotypes, such as compacted inflorescence and abnormal floral organs, during the reproductive stage (Figure 3A–E). The ox2 plants showed the most severe developmental defects and were used for further analysis. In ox2 plants, 52% of flowers (*n* = 180) developed six normal stamens at the correct position (Figure 3G), but the rest exhibited various defects in stamen development (Figure 3H–M). These defects included flowers with five stamens (18.9%, Figure 3H), more than four short stamens (11.1%, Figure 3I), and undeveloped or malformed anthers such as a part of the anther transformed into a petal-like structure (3.88%, Figure 3J–L). ox2 plants also developed flowers with retarded stamens and petals (13.33%, Figure 3M) and consistently produced significantly swollen or shrunken pollen (arrows in Figure 3 N,O). ox2 plants rarely produced seeds after self-pollination (Figure 3P).

Floral organ identity is mainly defined by ABC class genes, such as *AP1*, *AP2*, AP3, *PI*, and *AG*. In the inflorescences of *mac5a* and *MAC5A*-overexpressing plants, the expression of *AP1*, *AP2*, and *AG* changed at different levels (Figure 4A).

To gain insight the role of *MAC5A* in the GA pathway, we examined the expression levels of genes involved in GA biosynthesis and signaling pathways in inflorescences of *mac5a* mutants and *MAC5A*-overexpressing plants. The expression levels of *ENT-KAURENOIC ACID HYDROXYLASE 2* (*KAO2)*, *ENT-KAURENE OXIDASE* (*GA3*) and *GID1B*, *RGA*, and *RGL2* were higher in *mac5a* mutants and *MAC5A*-overexpressing plants (Figure 4B). In response to GA application, the dwarf phenotype of plants overexpressing *MAC5A* was partially rescued; however, their fertility was not elevated by GA inhibitor PAC treatment (Appendix A).

### 2.4. MAC5A Is Highly Expressed in the Stamen and Cooperates with RABBIT EARS (RBE) to Regulated Floral Organ Development

To examine the spatiotemporal expression pattern of *MAC5A*, a 2-kb genomic sequence of *MAC5A* native promoter was fused with the *β-glucuronidase* (*GUS*) reporter gene (*P_MAC5A_:GUS*). In the reproductive phase, the *P_MAC5A_:GUS* signal was low in the cauline leaf (Figure 5A), but high in the anther (Figure 5B,C,F) and the carpel just after pollination (Figure 5E,G,H). In siliques, the *P_MAC5A_:GUS* signal was highly detected in developing seeds (Figure 5I), whereas it disappeared in mature seeds (Figure 5J).

Using yeast two-hybrid (Y2H) screening, we found that MAC5A may be a potential interacting protein of RBE, which is a repressor of *AG* expression in *Arabidopsis* flowers [29,30,31]. Subsequently, their interaction was validated using a Y2H assay (Figure 6A). A bimolecular fluorescence complementation (BiFC) experiment was performed to further confirm their interaction (Figure 6B). The second whorl petals in *rbe* mutants are missing or replaced with filaments (Figure 6C) [29,30]. We constructed a *mac5a rbe* double mutants to detect the genetic interaction of *MAC5A* and *RBE*. The petal defect in *rbe* was at least partly rescued in the *mac5a rbe* double mutants (Figure 6C), suggesting that *MAC5A* is responsible for the second-whorl defect in *rbe* mutants. We found that the pollen grains in both *rbe* and *mac5a rbe* displayed an irregular and collapsed shape (Figure 6D), similar to those in the *mac5a* single mutant (Figure 1F).

## 3. Discussion

In this study, we unveiled the molecular role of MAC5A in floral organ development in *Arabidopsis*. Both the loss and gain of the *MAC5A* function frequently resulted in the malformation of floral organs, especially the stamens (Figure 1 and Figure 3), which indicates that an optimum expression of *MAC5A* is critical for the normal development of floral organs. *Arabidopsis* flowers consist of four whorls of organs: sepals, petals, stamens, and carpels; and their identity is defined by three classes of homeotic genes: A, B, and C. The class A genes *AP1* and *AP2* specify the identity of sepals; class A and B genes *AP3* and *PI* specify the identity of petals; class B and C genes specify the identity of stamens; and the class C gene *AG* specifies the identity of carpels. In the absence of the class C gene, the class A genes are active throughout the floral meristem. *AG* is required to repress the activity of class A genes in the third and fourth whorls [8,32]. The loss and gain of the *MAC5A* function affect the expression of the class A and C genes (Figure 4A), and *MAC5A* promoter activity was identified in the stamens (Figure 5). These results suggest that MAC5A may be involved in floral organ development by regulating class A and C genes.

*RBE*, which encodes a C2H2 zinc finger transcriptional factor, is specifically expressed in the petal primordia and acts as a second-whorl repressor of *AG* to limit its expression in flowers. The second whorl petals in *rbe* mutants are missing or replaced with filaments due to misexpression of *AG* in the second whorl [29,30]. RBE directly interacts with the promoter of *microRNA164 (miRNA164)* and negatively regulates its expression [31]. MAC5A is an RNA-binding protein required for microRNA (miRNA) biogenesis [33]. Indeed, *miRNA164* expression is decreased in the *mac5a* mutants [33]. We found that MAC5A may be a potential interacting protein of RBE, and the petal defect of *rbe* was at least partly rescued in the *mac5a rbe* double mutants (Figure 6), suggesting that *MAC5A* is responsible for the *rbe* second-whorl defects. In *Arabidopsis* flowers, *miRNA164* genes regulate the expression of *CUP-SHAPED COTYLEDON1* (*CUC1*) and *CUC2*, which encode critical transcriptional regulators involved in organ boundary specification [34,35]. RBE coordinately regulates the expression of *miRNA164* genes to impact the expression of *CUC1* and *CUC2*, which in turn regulate the events required for sepal and petal development [31]. On the other hand, the abundance of *miRNA172* is also reduced in *mac5a* mutants [33]. *miRNA172* mediates the repression of *AP2*, which is critical for stamen and carpel development [36,37]. We propose that *MAC5A* and *RBE* work together to regulate the class A and C genes through an miRNA-dependent pathway.

GAs play an essential role in plant growth, floral development, and late embryogenesis [5,38,39]. GA promotes floral organ development in *Arabidopsis* by blocking the functions of DELLA proteins [2]. After the binding of bioactive GA, the GA receptor GID1 binds to the DELLA protein to form a GA-GID1-DELLA complex [12,19,20]. Subsequently, the GA-GID1-DELLA complex is recognized by the SCFSLY1/GID2 ubiquitin E3 ligase complex for polyubiquitination, and the DELLA protein is then degraded [21,22]. Reduction in GA concentration promotes the accumulation of DELLA proteins that repress GA responses, whereas an increase in GA results in the degradation of DELLA proteins and activation of GA responses. In the inflorescence of *mac5a* mutants and *MAC5A-*overexpressing plants, the upregulated expression of GA biosynthesis and signaling genes (Figure 4B) may indicate that excessive levels of GA or exaggerated GA signaling lead to stamen deficiency in *mac5a* and *MAC5A-*overexpressing plants (Figure 1 and Figure 3), which also supported by the GA application exacerbating the deformation of the stamen in *mac5a* null mutants (Figure 2). The dwarf phenotype of ox2 plants was partially rescued by GA application; however, the fertility of plants overexpressing *MAC5A* was not significantly elevated by GA inhibitor PAC treatment (Appendix A), indicating that MAC5A also regulate stamen development in a GA-independent pathway. Five *DELLA* genes (*GAI*, *RGA*, *RGL1*, *RGL2*, and *RGL3*) are present in the *Arabidopsis* genome. The loss of function of *RGA* and *RGL2* is almost sufficient to compensate for floral organ deficiency in *ga1-3* mutants, which indicates that both *RGA* and *RGL2* are major repressors of GA response in this process [21]. We found that the expression levels of *RGA* and *RGL2* were significantly increased in *mac5a* and ox2 plants (Figure 4B), which suggested that *MAC5A* may play an important role in regulating the expression of *RGA* and *RGL2* during floral organ development. Taken together, these results indicate that optimal levels of *MAC5A* are necessary for the development of stamen. *MAC5A* gene may be involved in the GA pathway to regulate stamen development. Further analysis of the relationship between MAC5A and factors related to GA biosynthesis and signaling will provide insights into GA-mediated stamen development in plants.

## 4. Materials and Methods

### 4.1. Plants Materials and Growth Conditions

Col-0 plants were used as the wild type. *mac5a* (Salk_132881, *mac5a-1*) and *rbe* (CS6396, *rbe-2*) with a Col-0 genetic background were obtained from the *Arabidopsis* Information Source (TAIR, http://www.arabidopsis.org, 11 February 2022). Seeds were surface-sterilized and plated on a 0.5 × Murashige und Skoog basal salt (Wako, Osaka, Japan) medium containing 0.8% agar, 1% *w*/*v* sucrose, and 0.5 g/L MES (pH 5.8). After sowing, the plates were chilled at 4 °C for 72 h in dark and subsequently incubated at 22 °C under long-day photoperiods (LDs, 16/8 h light/dark). Light intensity was 120 μmol m^−2^ s^−1^. After ten days of growth, the plantlets were planted on rock wool cubes and grown with regularly supplied 0.2 × MS solution. To determine the effect of exogenous GA and PAC treatments on floral organs, GA_3_ and PAC were weekly applied to 20-day-old plants by spraying them with 50 μM or 100 μM GA_3_ (Sigma-Aldrich, St. Louis, MO, USA) or 20 μM PAC (Sigma-Aldrich) in 0.001% Triton X-100 or Triton X-100 alone.

### 4.2. Microscopy

Pollen viability was assessed using Alexander cytoplasmic staining [40] and observed using a stereomicroscope (SZX16; Olympus, Tokyo, Japan). Here, the cytoplasm of pollen was colored red, which indicated viable pollen grain. When the cytoplasm was absent, the cell wall of pollen grain can be stained, which indicated aborted pollen grain. The anther cannot be stained, which indicated invisible pollen grain. Progress in pollen development was determined using 4′,6-diamidino-2-phenylindole (DAPI; Sigma-Aldrich), and the DAPI signal was examined using a confocal microscope (LSM880; Zeiss, Oberkochen, Germany). Floral organs in fresh samples were observed using a benchtop scanning electron microscope (TM4000; Hitachi, Tokyo, Japan).

### 4.3. Preparation of Chimeric Constructs and Plant Transformation

To create the pBI121:*P_MAC5A_:GUS* construct, an approximately 2-kb long promoter sequence upstream of *MAC5A* was cloned from *Arabidopsis* genomic DNA using primer pairs with *Hind*III and *Sma*I sites. The CaMV 35S promoter of the pBI121 vector (Clontech, Mountain View, CA, USA) [41] was replaced with a *MAC5A* native promoter that was double-digested with *Hind*III/*Sma*I. *MAC5A* cDNA was obtained from ABRC and sequenced. Full-length *MAC5A* cDNA without a stop codon was cloned into a *Kpn*I/*Spe*I site of the vector pBS:*35S*:*GFP* [42], generating pBS:*35S*:*MAC5A-GFP*. The *MAC5A-GFP* fusion gene was cloned from pBS:*35S*:*MAC5A-GFP* using primer pairs with *Bam*HI and *Sac*I sites and cloned into the pBI121 vector to replace the *GUS* gene, generating pBI121:*35S*:*MAC5A:GFP*. Thus, *GFP* was fused to the C-terminus of *MAC5A*. pBS:*35S*:*GFP* was digested by *Xba*I and *Sac*I to obtain the open reading frame (ORF) of the *GFP* gene. *E**GFP* fragments were inserted into the *Xba*I/*Sac*I site of pBI121 to generate pBI121:*35S*:*GFP*. The primers are listed in Appendix A.

*Arabidopsis* Col-0 plants were transformed using *Agrobacterium tumefaciens* via the floral dip method [43]. We used the EHA105 *Agrobacterium* strain that harbored pBI121:*35S*:*GFP*, pBI121:*35S*:*MAC5A-GFP*, or pBI121:*P_MAC5A_:GUS*.

### 4.4. Histochemical GUS Assay

The GUS expression analysis was conducted as previously reported [44]. The chlorophyll of plants that were successfully transformed with the pBI121:*P_MAC5A_:GUS* vector was cleared with chilled 90% acetone. GUS activity was assessed by incubating plant tissues in 100 mM NaPO_4_ (pH 7.2), 5 mM 5-bromo-4-chloro-3-indolyl-D-glucuronide, 0.5 mM K_3_Fe(CN)_6_, 0.5 mM K_4_Fe(CN)_6_, and 0.25 Triton X-100 at 37 °C. Samples were cleared using 70% ethanol after staining.

### 4.5. Expression Analysis

Total RNA was isolated from inflorescences using an RNeasy Plant Mini Kit (QIAGEN, Dusseldorf, Germany), and cDNA was prepared using PrimeScript Reverse Transcriptase (TAKARA Bio, Shiga, Japan). Quantitative reverse-transcription PCR (qRT-PCR) was performed using a StepOne Real-Time PCR System (Applied Biosystems, Foster City, CA, USA) and SYBR Premix Ex Taq (TAKARA Bio). Actin (NM_112764) levels were used to normalize the expression patterns between samples. qRT-PCR primers used in the study are shown in Appendix A.

### 4.6. Y2H Assay

The ORFs of *MAC5A* and *RBE* were cloned into the *EcoR*I/*BamH*I I sites of pGADT7 and pGBKT7 vectors (Clontech, CA, USA), respectively. Bait and prey plasmids were cotransformed into yeast stain AH109 [45]. Empty vectors were used as negative controls. SD/-Leu/-Trp (DDO) was used to select for the bait and prey plasmids. SD/-Ade/-His/-Leu/-Trp (QDO) dropout supplement was used to confirm interactions. The yeast transformants were cultured with DDO at 30 °C for 24 h and subsequently harvested and diluted with distilled water. Then, 5 μL of the indicated dilution was dropped on DDO and QDO medium. Images were taken after the yeast cells were cultured for three to five days at 30 °C. The primers used for plasmid construction are shown in Appendix A.

### 4.7. Subcellular Localization and BiFC Assay

Full-length cDNAs of *MAC5A* and *RBE* without the stop codon were cloned into the *Kpn*I/*Spe*I site of the vector pBluescript II SK:35S:*GFP* (pBS:35S:*GFP*) [9], generating pBS:35S:*MAC5A-GFP* and pBS:35S:*RBE-GFP*. Vectors for the BiFC assay were constructed by replacing *GFP* in the vector pBS:35S:*GFP* with the N-terminus (154 amino acids) or C-terminus (80 amino acids) of *YFP*, generating pBS:35S:*nYFP* and pBS:35S:*cYFP*, respectively [42]. The ORFs of *MAC5A* and *RBE* without stop codons were amplified using a primer pair with the *Kpn*I and *Spe*I sites and cloned into pBS:35S:*cYFP* and pBS:35S:*nYFP*, respectively. The primers are shown in Appendix A. Plasmid DNA was introduced into onion epidermal cells using a bombardment system (Bio-Rad, Hercules, CA, USA, PDS-1000), and images were processed using Canvas X software (ACD Systems, Victoria, Canada).

## Figures and Tables

**Figure 1 ijms-23-02009-f001:**
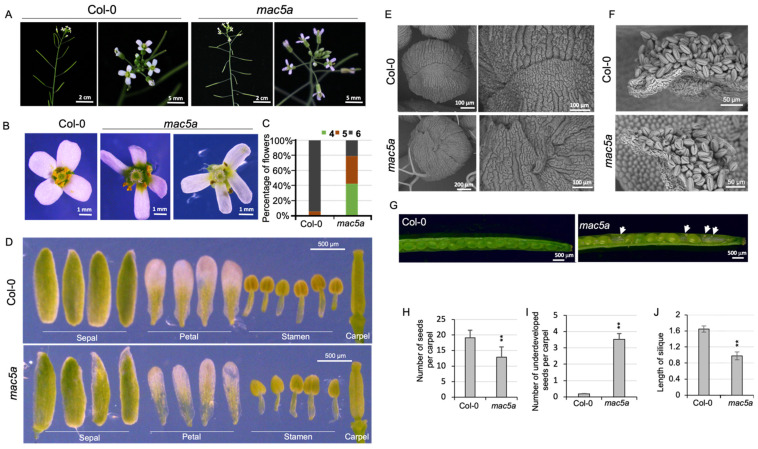
Phenotype of *mac5a* null mutants. (**A**) Representative inflorescences of wild-type Col-0 and *mac5a*. (**B**) Representative flowers of Col-0 and *mac5a*. (**C**) Percentages of flowers with 4, 5, and 6 stamens, respectively. (**D**) Dissected flower organs of Col-0 and *mac5a*. (**E**) Scanning electron micrograph of the outer epidermis of sepals of Col-0 and *mac5a*. The right panels showed the tip of the sepal enlarged from the left panels. (**F**) Scanning electron micrograph of pollen grains from Col-0 and *mac5a*. (**G**) Siliques of Col-0 and *mac5a*. White arrows indicate shrunken underdeveloped seeds. (**H**) Number of seeds in one carpel of Col-0 and *mac5a*. (**I**) Number of underdeveloped seeds in one carpel of Col-0 and *mac5a.* (**J**) Length of mature siliques of Col-0 and *mac5a*. Error bars represent S.E. *n* > 20. (Student’s *t*-test: ** *p* < 0.001).

**Figure 2 ijms-23-02009-f002:**
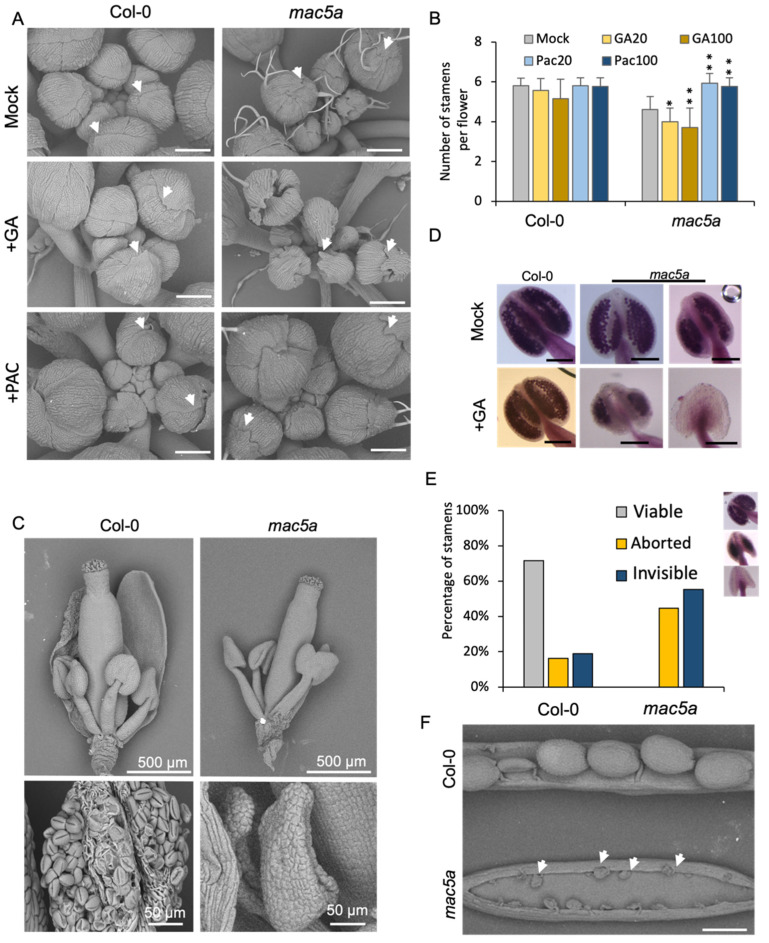
Phenotype of *mac5a* in response to GA and GA inhibitor PAC. (**A**) Representative inflorescences of wild-type Col-0 and *mac5a* after treatment with GA (50 μM) and PAC (20 μM). Scale bar = 200 μm. White arrows indicate the tip of the sepal. Error bars represent S.E. *n* > 20. (Student’s *t*-test: ** *p* < 0.001; * *p* < 0.05). Mock: control treatment without GA or PAC. (**B**) Number of stamens of Col-0 and *mac5a* flowers after treatment with GA and PAC; GA20: 20 μM GA_3_; GA100: 100 μM GA_3_; PAC 20: 20 μM PAC; PAC100: 100 μM PAC. Mock: control treatment without GA or PAC. (**C**) Scanning electron micrograph of flowers and anthers of Col-0 and *mac5a* after treatment with 100 μM GA_3_. (**D**) Alexander-stained anthers of Col-0 and *mac5a* after treatment with 100 μM GA_3_. Mock: control treatment without GA. Scale bar = 100 μm. (**E**) Percentage of anthers with different Alexander staining profiles in Col-0 and *mac5a* after treatment with 100 μM GA_3_. (**F**) Phenotype of mature siliques of Col-0 and *mac5a* after treatment with 100 μM GA_3_ treatment. White arrows indicate shrunken underdeveloped seeds. Scale bar = 500 μm.

**Figure 3 ijms-23-02009-f003:**
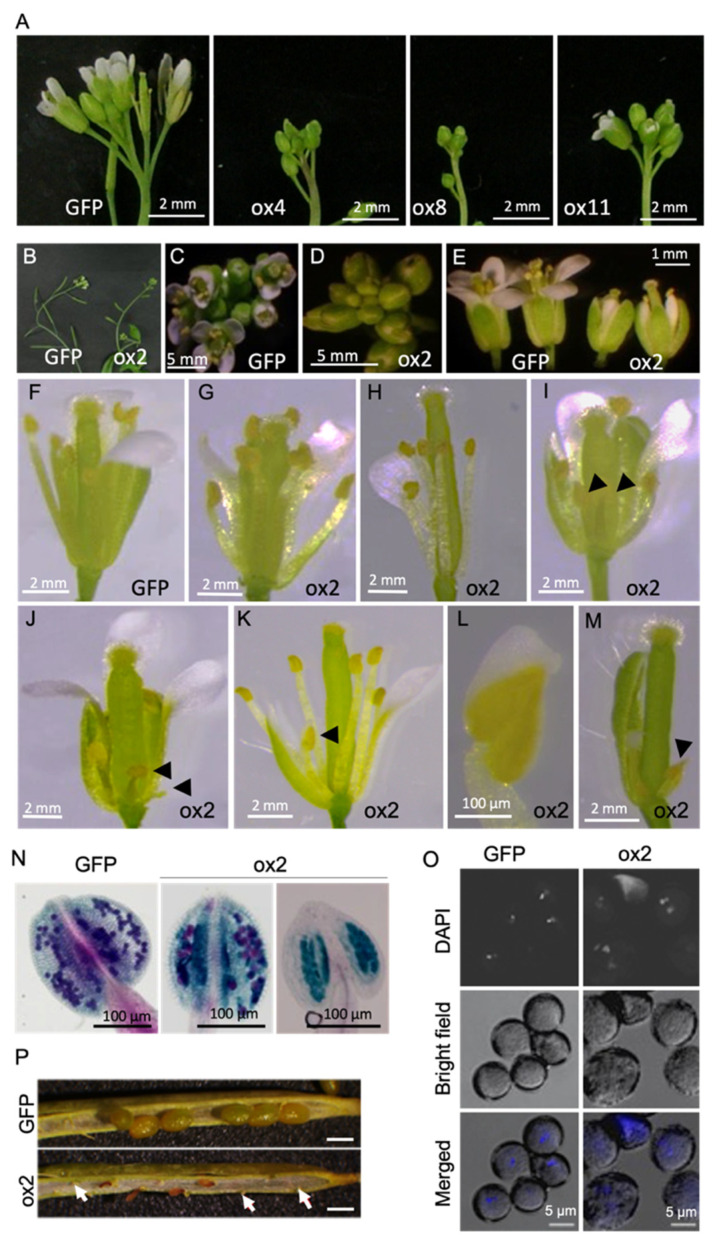
Phenotype of *MAC5A*-overexpressing plants. (**A**–**D**) Representative inflorescences of plants overexpressing *GFP* alone (GFP) or *GFP-MAC5A* (ox2, ox4, ox8, and ox11). (**E**–**M**) Representative flowers of GFP and ox2; black arrowheads indicate abnormal anthers. To examine the stamens, partial sepals and petals were removed. (**L**) The anther fused with petal was enlarged from panel (**K**). (**N**) Alexander-stained anthers from GFP and ox2. (**O**) DAPI-stained mature pollen grains of GFP and ox2. (**P**) Mature siliques of Col-0 and ox2. White arrows indicate shrunken underdeveloped seeds. Scale bar = 500 μm.

**Figure 4 ijms-23-02009-f004:**
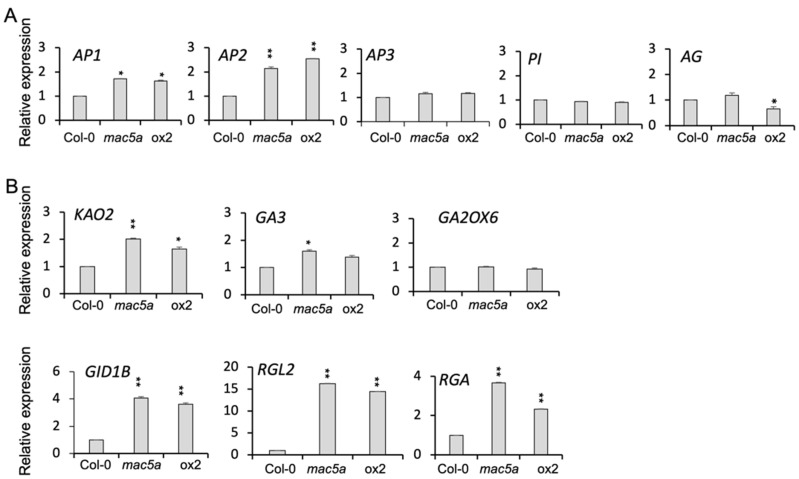
Expression of *ABC class* genes and *GA* biosynthesis and signaling genes in gain and loss of *MAC5A* function mutants. (**A**) Quantitative reverse-transcription PCR (qRT-PCR) validation of expression levels of *AP1*, *PI*, and *AG* genes in plants overexpressing *MAC5A*-GFP (ox2), *mac5a* null mutant, and Col-0. (**B**) qRT-PCR validation of expression levels of *GA* biosynthesis and signaling genes in Col-0, *mac5a*, and ox2. Error bars represent S.E. of three biological replicates. (Student’s *t*-test: ** *p* < 0.001; * *p* < 0.05).

**Figure 5 ijms-23-02009-f005:**
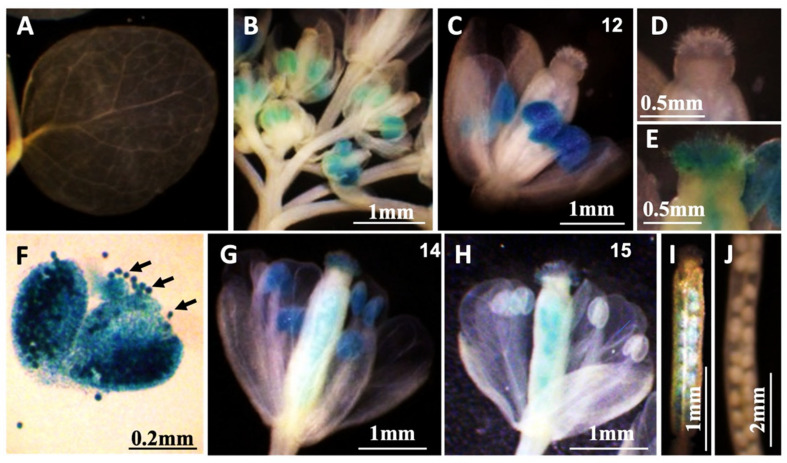
Expression of the β-glucuronidase reporter gene (*GUS*) driven by the native *MAC5A* promoter (*P_MAC5A_:GUS*) in flower. (**A**) Expression *of P_MAC5A_:GUS* in cauline leaf. (**B**) Expression *of P_MAC5A_:GUS* in inflorescences. (**C**,**G**,**H**) Flowers at different stages 12, 14, and 15. The numbers indicated in the right upper corner represented the flower stages. (**D**,**E**) The image of stigmas in panel (**C**,**G**) were enlarged in panel (**D**,**E**), respectively. (**F**) Anther. Black arrows indicated pollen grains. (**I**) Young silique. (**J**) Mature silique.

**Figure 6 ijms-23-02009-f006:**
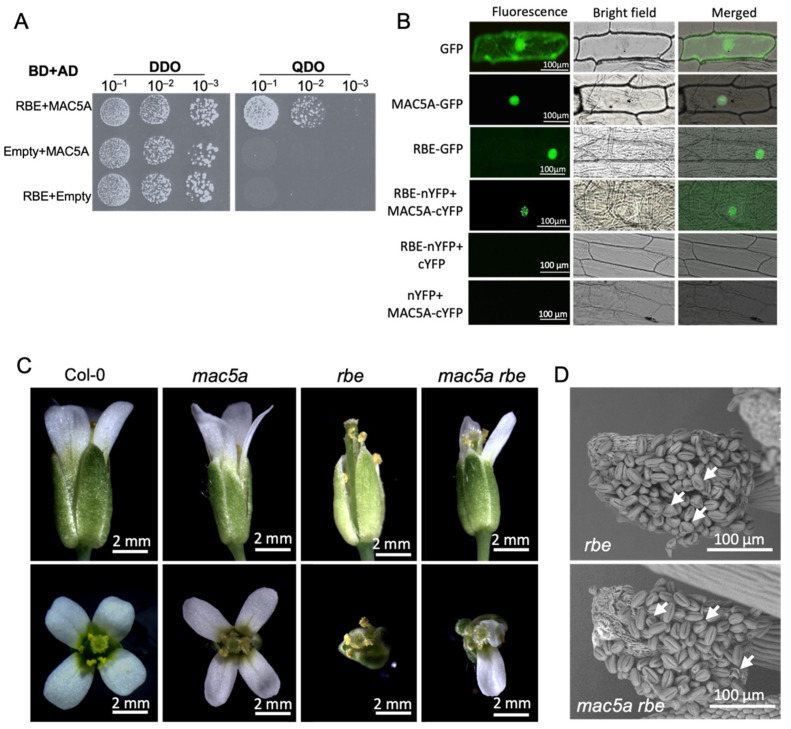
MAC5A interacted with RABBIT EARS (RBE) to regulate floral organ development. (**A**) Yeast two-hybrid analysis. In yeast cells that grew on quadruple dropout (QDO) medium, the reporter genes were activated when the bait (BD-RBE) interacted with the prey (AD-MAC5A). The combination of the pGBKT7 vector (empty) with AD-MAC5A and pGADT7 vector (empty) with BD-RBE are showed as the negative control. (**B**) Subcellular localization of GFP-tagged MAC5A (MAC5A-GFP) and RBE (RBE-GFP) and BiFC assay in onion epidermal cells. Fluorescence: GFP or YFP fluorescence. Merged: overlap of bright field and GFP signal. (**C**) Representative flowers of Col-0, *mac5a*, *rbe*, and *mac5a rbe*. (**D**) Representative anthers of *rbe* and *mac5a rbe*. White arrows indicate abnormal anthers.

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
