# Peer review of "RNA-Binding Protein MAC5A Is Required for Gibberellin-Regulated Stamen Development"

_ijms, 2022, doi:10.3390/ijms23042009_

Round 1
Reviewer 1 Report
The manuscript is much improved when compared to its earlier version. There are minor errors in language or grammar that can be corrected.
line 234; What is EBE?
Reviewer 2 Report
The manuscript entitled with “RNA-binding protein MAC5A is required for gibberellin-regulated stamen development” was studied by Liu et al. Although the functional roles of MAC5A were previously analyzed by Monaghan et al. (2010) and Li et al. (2020), the authors have provided the additional information of MAC5A associated with a possible interaction partner (RBE) and a development of stamen on the GA signaling pathway. The results looks interesting and I suggest several comments below to improve the manuscript.
- In the figure legends, english “shown are the means…” need to be rectified with a proper expression.
- In the Figure 2E, the anther of “Aborted” showed that the decreased number of pollen grains in anther sac. It looks like there was no difference in pollen grain activity between the pictures of viable and aborted. Please clarify them.
- In line 143, T2 should be with a subscript.
- In line 153, Ox2 should be with a small letter.
- The Figure 3 need to be repositioned below the subheading and the Figure 5 is better to put in Supplemental Figure.
- In Figure 7A, the resolution of picture is unclear. Please replace it with the high quality of the picture with a dilution experiment.
- In Figure 7B, please include a negative control of RBE-nYFP and MAC5A-cYFP for the BiFC experiment.
- In the material and method, please add a citation for the Histochemical GUS assay.
- In the material and method, please supplement more explanation of Y2H experiment such as a yeast strain, cell concentrations, citations etc.
- Please check the reference format of Journal.
Reviewer 3 Report
The manuscript by Liu et al reported MAC5A plays a key role in gibberellin-regulated stamen development through genetic studies. They further uncovered the mechanism that MAC5A mediates the normal development of stamens through the GA signaling pathway by interacting with RBE. The findings here have far-reaching significance to the field. The manuscript is clearly written, and I have only a few suggestions.
Minor issues:
- Fig1 E and F, please label all the scale bars.
- Fig4, what are the protein expression levels of those ABC class genes, especially those increase significantly in mRNA level?
- Both MAC5A and RBE are RNA binding proteins. Therefore, RNA may regulate the interaction between MAC5A and RBE. Both Y2H and BiFC assays can not determine whether the interaction is directly or not. Can the authors verify the interaction using other methods like GST-pull down or Isothermal titration calorimetry?
Round 2
Reviewer 2 Report
The authors addressed all my points and concerns.
Author Response
Dear Reviewer, We appreciate your positive comments. Best regards, HuaThis manuscript is a resubmission of an earlier submission. The following is a list of the peer review reports and author responses from that submission.
Round 1
Reviewer 1 Report
The study is scientific merits, but it needs to be significantly improved before publication:
- The introduction is too brief.
- Add scale bars in Figure 1-B, D, E (enlarged panels), G.
- Figure 1-D: indicate the organ names
- Line 99: should be resulting in
- Figure 2A: please arrow to indicate tip of sepal
- Figure 2B: add title for y-axis; use letters or * to indicate statistical differences.
- Figure 2E: add title for y-axis
- What was the justification for using GFP as the reference for qRT-PCR?
- Figure 5: why are the main title and subtitle (A) the same?
- Line 136-139: the result of “Additionally, the dwarf phenotype of 136 ox2 plants was partially rescued of by GA application (Figure 6).” has nothing to do with “Taken together, we hypothesized that MAC5A specifically expressed in the stamen regulates floral organs development by affecting the expression of ABC class genes.”
- Line 153: change “A alone” to “A class alone”
- Line 153: correct grammars in “A and B specifies petals, B and C specifies stamens”
- Line 160-161: “These results suggested that MAC5A is a novel factor in floral organ development and MAC5A may act in concert with A class gene AP1 to properly activate C class gene AG.”:
- The discussion is not really a discussion. Basically it just repeats the results.
- How was the concentration of exogenous application of GA3 and PAC determined? Would a different concentration have changes the results in the current study?
- Line 196: did you mean genomic DNA?
- Line 198: how was SpeI site introduced? There is no SpeI site in pBI121 plasmid
Reviewer 2 Report
Title: RNA binding protein MAC5A is required for GA-regulated sta-2 men development.
In this study, the authors show involvement of MOS4-Associated Complex 5A (MAC5A), an RNA- binding motif containing protein, in the development of sepal, petal and stamen by affecting the expression of homeotic genes AP1, PI and AG.
They report that the loss-of-function and the overexpression of MAC5A both resulted in malformation of stamen and reduction of seed-set. They further show that MAC5A is regulated by GA signaling pathway.
A very intresting study supported by changes in plant phenotypes, however the results obtained have not been described properly. The discussion section is weak and many points have not been addressed.
- Fig 1 compare WT and mac knockout (KO) phenotypes. Authors should indicate that these pictures are representing the entire KO population like in panel B the mac KO flowers show variation Panel G shows variation in seed set but is true in all cases. Do WT siliques always have 100% seed set? It is best to support some observations in a table mentioning the average values.
Also individual panels can be properly labelled, especially in panel C the legends to graph axis need to be added and in panel D the various organs should be labelled.
- The related writeup needs to be described and supported well. The root and shoot phenotypes have been described but pictorial comparions are not present. Similarly the in text descriptions of each panel do not correlate well. Do the pics shown in panel D and E taken at the same stage? Does panel F indicate adifference in dehisence as well?
- Fig 2 doenat support the authors statement that “GA treatment enhanced the abnormalities in floral organs”. Panel B shows a decrease in stamen number but what about morphology? Panel D should swhow three stages discussed in panel E. Can a picture be provided to show the effect of GA?
- The effect of mac overexpression (OX) on flower phenotype needs to be carefully described. From the pic it appears to affect the floral arrangement and growth on the whole? Are all pictures captured at the same stage?
- What will be the effect of GA on the OX phenotype?
- In Fig 4. Please include the levels in KO plants
- Why are the OX and KO lines behaving in a similar manner? Have the authors ruled out silencing?
- Discussion is too short and needs to be substantially expanded
- Related work on MOS4 has not been cited
Round 2
Reviewer 1 Report
The minor items were adequately addressed. The discussion is still too shallow. It needs major improvement. Also, what being studied is the gene, not protein. So the authors need to pay attention to the way they represented the results.
Table S1: please indicate 5’ and 3’. The authors mentioned actin was used in qRT-PCR, but its primer sequences are not in Table S1.
qRT-PCR: what method was used for the expression analysis?
Author Response
Reviewer 1The study is scientific merits, but it needs to be significantly improved before publication:
Response:we appreciate your critical and positive comments. The manuscript was
carefully revised.
• The introduction is too brief.
Response:we introduce ABC model and information related to MAC in detail.
• Add scale bars in Figure 1-B, D, E (enlarged panels), G.
Response: we apologize for our mistakes. Scale bars were added.
• Figure 1-D: indicate the organ names
Response: added.
• Line 99: should be resulting in
Response: revised.
• Figure 2A: please arrow to indicate tip of sepal
Response: added.
• Figure 2B: add title for y-axis; use letters or * to indicate statistical differences.
Response: added.
• Figure 2E: add title for y-axis
Response: added.
• What was the justification for using GFP as the reference for qRT-PCR?
Response: plants overexpressing GFP alone (GFP) were used as a control.
Arabidopsis Actin gene (NM_112764) was used as a reference to normalize the
expression.
• Figure 5: why are the main title and subtitle (A) the same?
Response: revised.
• Line 136-139: the result of “Additionally, the dwarf phenotype of 136 ox2 plants was
partially rescued of by GA application (Figure 6).” has nothing to do with “Taken
together, we hypothesized that MAC5A specifically expressed in the stamen
regulates floral organs development by affecting the expression of ABC class genes.”
Response: revised as following:
These results suggested that MAC5A specifically expressed in the stamen regulates floral
organs development by affecting the expression of ABC class genes. Additionally, the
dwarf phenotype of ox2 plants was partially rescued of by GA application (Fig. 6). Taken
together with that GA application led to reduction of the stamen number and retarded
anther development in mac5a (Fig. 2), we hypothesized MAC5A is involved in the GA
signaling pathway to regulate stamen development (line 165-171 in the revised version).
• Line 153: change “A alone” to “A class alone”
Response: revised.
• Line 153: correct grammars in “A and B specifies petals, B and C specifies stamens”
Response: revised.
• Line 160-161: “These results suggested that MAC5A is a novel factor in floral organ
development and MAC5A may act in concert with A class gene AP1 to properly
activate C class gene AG.”:
• The discussion is not really a discussion. Basically it just repeats the results.
Response: discussion section was revised and we promote a hypothesis as following:
These results suggested that MAC5A may be a novel factor in floral organ development and
MAC5A may act in regulating C class gene AG to repress A class gene AP1 (line 197-198 in
the revised version).
• How was the concentration of exogenous application of GA3 and PAC determined?
Would a different concentration have changes the results in the current study?
Response: we tried different concentrations of GA3 and PAC (2, 5, 10, 20, 50, 100
μM). In our case, GA3 application causes dose-dependent effect on the fertility of
mac5a null mutant, alternatively, higher concentration of GA3 leads to lower fertility of
mac5a. Lower concentration (2-20 μM) of PAC rescues the abnormal phenotype of
mac5a, but high concentration (50-100 μM) of PAC inhibits the growth and
development of both Col-0 and mac5a.
• Line 196: did you mean genomic DNA?
Response: Yes. It was revised as following:
To generate pBI121:PMAC5A:GUS construct, an approximately 2kb promoter sequence
upstream of MAC5A gene was cloned from an Arabidopsis genomic DNA by using primer
pair with HindIII and SmaI sites. (line 245 in the revised version).
Line 198: how was SpeI site introduced? There is no SpeI site in pBI121 plasmid
Response: we apologize for our mistakes. HindIII site was used. It was revised as
following:
an approximately 2kb promoter sequence upstream of MAC5A gene was cloned from an
Arabidopsis genomic DNA by using primer pair with HindIII and SmaI sites. The CaMV
35S promoter of pBI121 vector (Clontech) (Bevan, 1984) was replaced with MAC5A
native promoter double-digested with HindIII/SmaI (line 246-248 in the revised version)
Reviewer 2 Report
The manuscript is much improved now and figures are better labelled. Please get the manuscript see by an english language expert, as grammatical corrections are required at several places. few examples are mentioned below and there are many more like correct use of tenses as in had with singular etc.
- Abstract: "Gibberellin (GA) signaling is involved into floral organ..." Replace "into" with "in"
- ".....we found that.." replace with "we show that"
- "...involved into.." replace with "involve in"
- title 2.2 "deficient" replace with "deficiency"
